# Orthonormalising gradients improves neural network optimisation

## Abstract

The optimisation of neural networks can be improved by orthogonalising the gradients before the optimisation step, ensuring the diversification of the learned intermediate representations. We orthonormalise the gradients of a layer's components/filters with respect to each other to separate out the latent features. Our method of orthogonalisation allows the weights to be used more flexibly, in contrast to restricting the weights to an orthogonal sub-space. We tested this method on image classification, ImageNet and CIFAR-10, and on the semi-supervised learning BarlowTwins, obtaining both better accuracy than SGD with fine-tuning and better accuracy for naïvely chosen hyper-parameters.

## 1 Introduction

Neural network layers are made up of several identical, but differently parametrised, components, *e.g.* filters in a convolutional layer, or heads in a multi-headed attention layer. Layers consist of several components so that they can provide a diverse set of intermediary representations to the next layer, however, there is no constraint or bias, other than the implicit bias from the cost function, to learning different parametrisations. We introduce this diversification bias in the form of orthogonalised gradients and find a resultant speed-up in learning and improved performance, see Figure 1.

Our novel contributions include this new optimisation method, thorough testing on CIFAR-10 and ImageNet, additional testing on a semi-supervised learning method, and experiments to support our hypothesis.

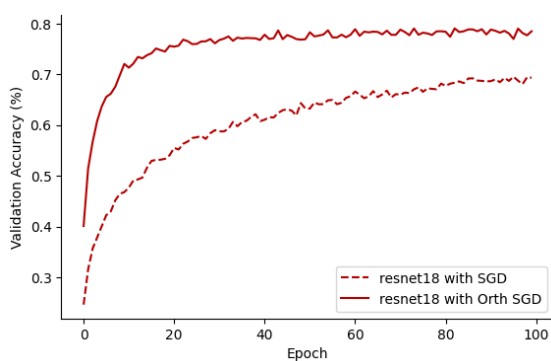

Figure 1: An example of the speed-up obtained by orthogonalising the gradients on CIFAR-10.

In Section 2 we detail the method and results to give an understanding of how this method works and its capabilities. Then, in Section 4, we provide experimental justifications and supporting experiments for this method along with finer details of the implementation and limitations.

## 2 Overview of new method and results

### 2.1 Related works

Gradient orthogonalisation has been explored in the domain of multi-task learning (Yu et al., 2020) to keep the different tasks separate and relevant. However in this work we focus on orthogonalisation for improving single task performance.

Weight orthogonalisation has been extensively explored with both empirical (Bansal et al., 2018; Jia et al., 2017) and theoretical (Jia et al., 2019) justifications. However, modifying the weights during training is unstable, and, in addition, it limits the weights to a tiny subspace. Deep learning is know to work well despite the immense size of the weight space, and as such we do not view this as an advantage. Xie et al. (2017) obtain improved performance over Stochastic Gradient Descent (SGD) via weight orthogonalisation and allows them to train very deep networks, we aim to achieve the same results while being more flexible with model and optimisation method choice. We do this by orthogonalising the gradients before they are used by an optimisation method rather than modifying the weights themselves. This, in effect, biases the training towards learning orthogonal representations and we show this to be the case; in many areas of deep learning, introducing a bias instead of a hard constraint is preferred.

## 2.2 Orthogonalising Gradients

Given a neural network, $f$, with $L$ layers made from components, $c$,

$$f = \circ_{i=1}^{L}(f_i), \tag{1}$$
$$f_l(x) = [c_{l1}(x), c_{l2}(x), \ldots, c_{lN_l}(x)], \tag{2}$$

where $\circ$ is the composition operator, $N_l$ is the number of components in layer $l$, $c_l : \mathbb{R}^{S_{l-1} \times N_{l-1}} \to \mathbb{R}^{S_l}$ is a parametrised function and $c_{l_i}$ denotes $c_l$ parametrised with $\theta_{l_i} \in \mathbb{R}^{P_l}$ giving $f_l : \mathbb{R}^{S_{l-1} \times N_{l-1}} \to \mathbb{R}^{S_l \times N_l}$ parametrised by $\theta_l \in \mathbb{R}^{P_l \times N_l}$.

Let

$$G_l = [\nabla c_{l1}, \nabla c_{l2}, \ldots, \nabla c_{lN_l}], \tag{3}$$

be the $P_l \times N_l$ matrix of the components' gradients.

Then the nearest orthonormal matrix, *i.e.* the orthonormal matrix, $O_l$, that minimises the Frobenius norm of its difference from $G_l$

$$\min_{O_l} \|O_l - G_l\| \quad \text{subject to } \forall i,j : \langle O_{l_i}, O_{l_j} \rangle = \delta_{ij},$$

where $\delta_{ij}$ is the Kronecker delta function, is the product of the left and right singular vector matricies from the Singular Value Decomposition (SVD) of $G_l$ (Trefethen & Bau III, 1997),

$$G_l = U_l \Sigma_l V_l^{\mathsf{T}}, \tag{4}$$
$$O_l = U_l V_l^{\mathsf{T}}. \tag{5}$$

Thus, we can adjust a first-order gradient descent method, such as Stochastic Gradient Descent with Momentum (SGDM) (Polyak, 1964), to make steps where the components are pushed in orthogonal directions,

$$v_l^{(t+1)} = \gamma v_l^{(t)} + \eta O_l^{(t)}, \text{ and} \tag{6}$$
$$\theta_l^{(t+1)} = \theta_l^{(t)} - v_l^{(t+1)}, \tag{7}$$

where $v_l$ is the velocity matrix, $t \in \mathbb{Z}^{0+}$ is the time, $\gamma$ is the momentum decay term, and $\eta$ is the step size. We call this method Orthogonal Stochastic Gradient Descent with Momentum (Orthogonal-SGDM). This modification can clearly be applied to any first-order optimisation algorithm by replacing the gradients with $O_l^{(t)}$ before the calculation of the next iterate.

Code for creating orthogonal optimisers in PyTorch is provided at `https://anonymous.4open.science/r/Orthogonal-Optimisers`. And code for the experiments in this work is provided at `https://anonymous.4open.science/r/Orthogonalised-Gradients`

Table 1: Test loss and accuracy of a resnet20 trained with SGDM and Orthogonal-SGDM across five runs on CIFAR-10; ‡ is as in He et al. (2015): batch size of 128, learning-rate of 0.1, momentum of 0.9, weight-decay of $10^{-4}$, and a learning rate schedule of $\times 0.1$ at epochs 100, 150 for 200 epochs; † is the best hyper-parameters found for Orthogonal-SGDM via Bayesian optimisation: batch size of 1024, learning-rate of $5 \times 10^{-2}$, momentum of 0.85, weight-decay of $10^{-2}$, and a learning rate schedule of $\times 0.1$ at epochs 100, 150 for 200 epochs.

|                          | Test Loss           | Test Acc (%)       |
| ------------------------ | ------------------- | ------------------ |
| SGDM (He et al., 2015) ‡ | —                   | 91.25              |
| SGDM‡                    | $0.4053 \pm 0.0054$ | $91.17 \pm 0.28$   |
| Orthogonal-SGDM‡         | $0.4231 \pm 0.0043$ | $90.18 \pm 0.30$   |
| SGDM†                    | $0.3062 \pm 0.0036$ | $91.45 \pm 0.09$   |
| Orthogonal-SGDM†         | $0.3553 \pm 0.0045$ | **92.08** $\pm 0.05$ |

## 3 Results

### 3.1 CIFAR-10

#### 3.1.1 Bettering ResNet's performance

First we test the efficacy of our method in reproducing the results of a resnet20 from the original ResNet paper (He et al., 2015) on CIFAR-10 (Krizhevsky et al., 2009). In this work we are concerned with just the full pre-activation version of residual networks and the original identity mappings (He et al., 2016) for shortcut connexions.

To do this we use the same hyper-parameters as the original, which have been tuned to benefit SGDM, to train using Orthogonal-SGDM. Table 1 shows that for these runs (marked with ‡) we can reproduce the results of the original paper and that Orthogonal-SGDM obtains similar, if not quite so stellar, performance.

Next, we use Bayesian hyper-parameter optimisation on the learning-rate, momentum, and weight-decay to tune the hyper-parameters for Orthogonal-SGDM and show that it outperforms both the original SGDM hyper-parameters and SGDM with the same hyper-parameters († in Table 1). While new architectures have lead to better performing resnet-models *e.g.* (He et al., 2016; Guo et al., 2016; Wu et al., 2019), this is, to the best of the authors' knowledge, the state-of-the-art for the original model.

Overall, this experiment demonstrates that Orthogonal-SGDM can be used as a good default optimiser in new projects, and that it can be used as a drop-in replacement for SGDM in existing projects to obtain a small gain in performance.

#### 3.1.2 Untuned optimisation & Adam

We compare our method to the Adam optimiser (Kingma & Ba, 2014) for different learning-rate and momentum terms.

Adam has found its place as a reliable optimiser that works over a wide variety of hyper-parameter sets, yielding consistent performance with little fine-tuning needed, such that most of the performance is gained under most hyper-parameter sets. In this experiment we show that our adaption allows Orthogonal-SGDM to outperform Adam on all but two hyper-parameter sets, Table 2, and where Adam is better, it is in turn beaten by Orthogonal-Adam — applying our modification to Adam itself. In addition, Orthogonal-Adam is superior at high learning rates where Adam suffers from blow-ups. See Figures 12 and 13 for the training plots.

All first order method have to store $2n$ parameters, where $n$ is the size of the model, the model itself and the calculated gradients; however, when an optimiser uses an extra buffer we need an additional $n$ memory. All oft-used optimisers also use an extra buffer to store either some cumulation of previous gradients (usually called the momentum buffer), *e.g.* SGDM, taking $3n$-memory, or a buffer of some cumulation of the squared

Table 2: Test accuracy across a suite of hyper-parameter sets on CIFAR-10 on a resnet20 for 100 epochs using a batch size of 1024 and a weight-decay of $5 \times 10^{-4}$, standard error across five runs. For Adam $\beta_2 = 0.99$,

| | | SGDM | | Adam | |
|---|---|---|---|---|---|
| LR | Mom | Original | Orthogonal | Original | Orthogonal |
| $10^{-1}$ | 0.95 | $83.59_{\pm 2.09}$ | $\mathbf{85.58}_{\pm 0.98}$ | $38.95_{\pm 8.95}$ | $76.84_{\pm 1.53}$ |
| $10^{-2}$ | 0.95 | $82.66_{\pm 1.02}$ | $\mathbf{87.72}_{\pm 0.44}$ | $74.23_{\pm 2.38}$ | $86.48_{\pm 0.17}$ |
| $10^{-3}$ | 0.95 | $66.59_{\pm 0.44}$ | $\mathbf{85.88}_{\pm 0.33}$ | $83.08_{\pm 0.76}$ | $85.12_{\pm 0.06}$ |
| $10^{-1}$ | 0.9 | $82.52_{\pm 1.16}$ | $\mathbf{85.06}_{\pm 0.47}$ | $28.26_{\pm 7.16}$ | $73.62_{\pm 2.96}$ |
| $10^{-2}$ | 0.9 | $79.96_{\pm 0.48}$ | $\mathbf{87.44}_{\pm 0.25}$ | $73.46_{\pm 1.19}$ | $85.26_{\pm 0.38}$ |
| $10^{-3}$ | 0.9 | $60.69_{\pm 0.18}$ | $84.67_{\pm 0.21}$ | $83.16_{\pm 0.66}$ | $\mathbf{85.25}_{\pm 0.31}$ |
| $10^{-1}$ | 0.8 | $84.16_{\pm 0.43}$ | $\mathbf{86.01}_{\pm 0.74}$ | $27.50_{\pm 6.76}$ | $71.88_{\pm 4.25}$ |
| $10^{-2}$ | 0.8 | $77.42_{\pm 0.98}$ | $\mathbf{87.18}_{\pm 0.12}$ | $72.60_{\pm 1.76}$ | $86.75_{\pm 0.26}$ |
| $10^{-3}$ | 0.8 | $53.21_{\pm 0.43}$ | $82.95_{\pm 0.40}$ | $80.89_{\pm 1.93}$ | $\mathbf{85.52}_{\pm 0.29}$ |
| $10^{-1}$ | 0.5 | $80.08_{\pm 0.36}$ | $\mathbf{87.37}_{\pm 0.18}$ | $18.39_{\pm 4.84}$ | $72.10_{\pm 2.83}$ |
| $10^{-2}$ | 0.5 | $68.64_{\pm 1.05}$ | $\mathbf{86.05}_{\pm 0.10}$ | $71.62_{\pm 1.95}$ | $84.22_{\pm 0.69}$ |
| $10^{-3}$ | 0.5 | $43.51_{\pm 1.02}$ | $78.68_{\pm 0.77}$ | $81.67_{\pm 1.05}$ | $\mathbf{84.21}_{\pm 0.43}$ |
| $10^{-1}$ | 0 | $74.42_{\pm 0.63}$ | $\mathbf{83.80}_{\pm 0.32}$ | $10.00_{\pm 0.00}$ | $55.14_{\pm 6.97}$ |
| $10^{-2}$ | 0 | $53.42_{\pm 0.20}$ | $\mathbf{82.18}_{\pm 0.55}$ | $60.47_{\pm 3.70}$ | $80.50_{\pm 1.04}$ |
| $10^{-3}$ | 0 | $32.41_{\pm 0.57}$ | $63.25_{\pm 0.57}$ | $76.76_{\pm 2.09}$ | $\mathbf{83.45}_{\pm 0.40}$ |

gradients, *e.g.* Adaptive Gradient method (AdaGrad) or Root Mean Square Propogation (RMSProp), again requiring $3n$-memory; in some cases the optimiser stores both these buffers *e.g.* Adam and Weight-decayed Adam (AdamW) using $4n$-memory. Note how Orthogonal-SGDM and Orthogonal-Adam are able to obtain good results with 0 momentum, Table 2, which means that this training run can be implemented using only $2n$-memory, this would be useful for training huge models – for example foundational language models – where storing the extra buffer would either use a prohibitive amount of memory or force the use of a smaller model.

Large models trained with large datasets makes it infeasible to tune the hyper-parameters, therefore, we need an optimiser that works well on a wide range of hyper-parameters, so that it can train with the researcher's "best guess". Table 2 proves that Orthogonal-SGDM holds this property even more that Adam.

### 3.1.3 That Orthogonal-SGDM speeds up learning

Having shown that Orthogonal-SGDM can achieve state-of-the-art results and better results across the range of hyper-parameters, we now aim to show that it significantly speeds up and improves learning, *i.e.* the performance of Orthogonal-SGDM is equal to or better than SGDM at all points in training.

We trained a suite of models on the CIFAR-10 (Krizhevsky et al., 2009) data set with a mini-batch size of 1024, learning-rate of $10^{-2}$, momentum of 0.9, and a weight decay of $5 \times 10^{-4}$ for 100 epochs. We then repeated this using Orthogonal-SGDM instead of SGDM and plot the results in Figures 2 and 3 and Table 3.

Orthogonal-SGDM is more efficient and achieves better test accuracy than SGDM for every model we trained in this experiment. The validation curves follow the training curves, Figures 4 and 5, and have the same patterns, this means that Orthogonal-SGDM exhibits the same generalisation performance as SGDM. More importantly though, we can see that the model learns much faster at the beginning of training, as shown by Figure 2, this means that we do not need as many epochs to get to a well-performing network. This is especially good in light of the large data sets that new models are being trained on, where they are trained for only a few epochs, or even less (Brown et al., 2020). These results are not state-of-the-art, *cf.* Table 1, but

---

[1]As described in Appendix A.1

[2]Model same as in He et al. (2015)

[3]From `https://pytorch.org/vision/0.9/models.html`

Table 3: Test loss and accuracy across a suite of models on CIFAR-10 comparing normal SGDM with Orthogonal-SGDM, standard error across five runs.

| | Test Loss | | Test Accuracy (%) | |
|---|---|---|---|---|
| | SGDM | Orthogonal-SGDM | SGDM | Orthogonal-SGDM |
| BasicCNN[1] | $0.7603 \pm 0.0061$ | $0.6808 \pm 0.0038$ | $73.60 \pm 0.19$ | $\mathbf{76.67} \pm 0.10$ |
| resnet20[2] | $0.6728 \pm 0.0301$ | $0.6766 \pm 0.0155$ | $79.14 \pm 0.62$ | $\mathbf{87.12} \pm 0.12$ |
| resnet44[2] | $0.7000 \pm 0.0166$ | $0.7600 \pm 0.0299$ | $79.81 \pm 0.37$ | $\mathbf{88.12} \pm 0.20$ |
| resnet18[3] | $0.9656 \pm 0.0104$ | $0.8427 \pm 0.0121$ | $77.01 \pm 0.21$ | $\mathbf{84.68} \pm 0.12$ |
| resnet34[3] | $1.0468 \pm 0.0134$ | $0.7087 \pm 0.0165$ | $75.86 \pm 0.26$ | $\mathbf{85.42} \pm 0.33$ |
| resnet50[3] | $1.2304 \pm 0.0462$ | $0.6797 \pm 0.0235$ | $67.99 \pm 0.73$ | $\mathbf{86.51} \pm 0.12$ |
| densenet121[3] | $1.0027 \pm 0.0132$ | $0.8669 \pm 0.0132$ | $75.26 \pm 0.30$ | $\mathbf{84.34} \pm 0.15$ |
| densenet161[3] | $1.1399 \pm 0.0096$ | $1.1688 \pm 0.1960$ | $75.81 \pm 0.20$ | $\mathbf{85.51} \pm 0.19$ |
| resnext50_32x4d[3] | $1.2470 \pm 0.0254$ | $0.6669 \pm 0.0223$ | $68.73 \pm 0.30$ | $\mathbf{86.37} \pm 0.24$ |
| wide_resnet50_2[3] | $1.4141 \pm 0.0337$ | $0.7018 \pm 0.0091$ | $69.42 \pm 0.33$ | $\mathbf{87.30} \pm 0.12$ |

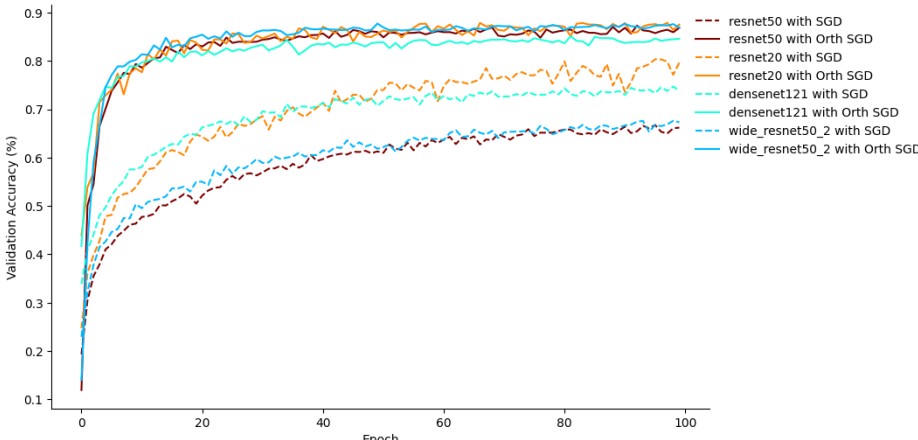

Figure 2: Validation accuracy from one run of SGDM vs Orthogonal-SGDM for a selection of models. Full plot in Appendix B. Best viewed in colour.

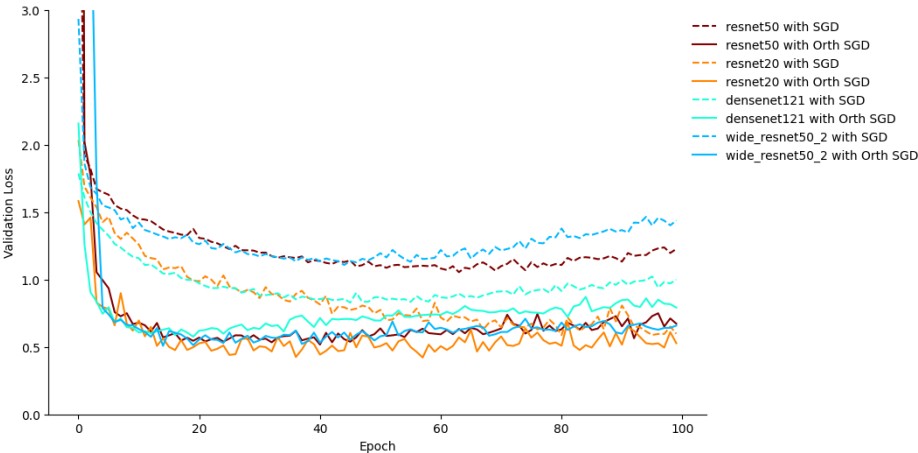

Figure 3: Validation losses from one run of SGDM vs Orthogonal-SGDM for a selection of models. Full plot in Appendix B. Best viewed in colour.

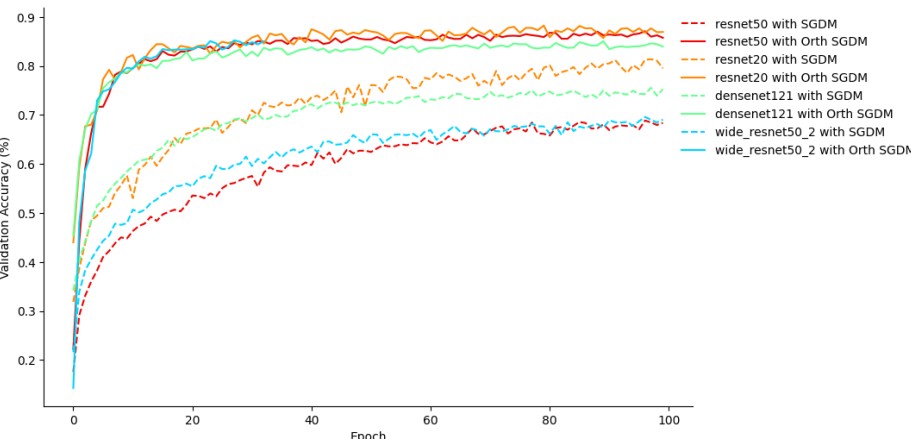

Figure 4: Train accuracy from one run of SGDM vs Orthogonal-SGDM for a selection of models. Best viewed in colour.

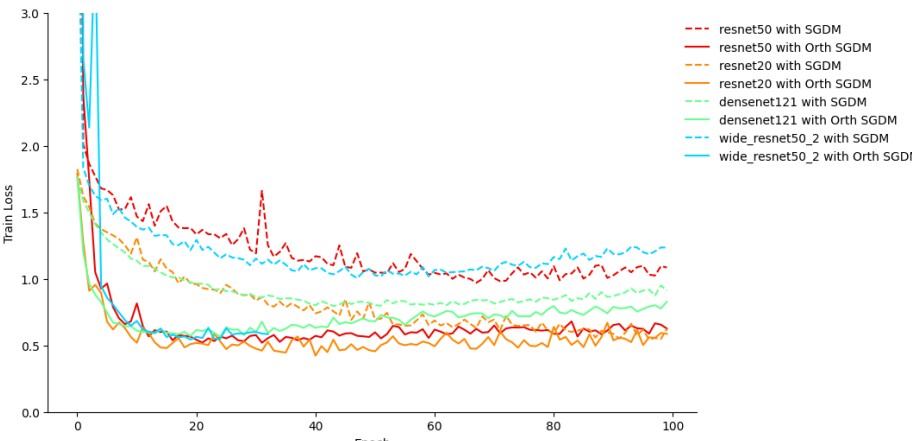

Figure 5: Train losses from one run of SGDM vs Orthogonal-SGDM for a selection of models. Best viewed in colour.

they exemplify the results of a starting point for real world problems and show case how Orthogonal-SGDM gets reasonably close to state-of-the-art when SGDM cannot.

The performance of the residual networks designed for ImageNet (Deng et al., 2009) resnet(18, 34, 50) under SGDM get worse with the increasing model's size. The original ResNet authors, He et al. (2015), note that unnecessarily large networks may over-fit on a small data set such as CIFAR-10. However, with Orthogonal-SGDM, these models do not suffer from this over-parametrisation problem and even slightly improve in performance as the models get bigger, in clear contrast to SGDM. This agnosticism to over-parametrisation helps alleviate the need for the practitioner to tune a model's architecture to the task at hand to achieve a reasonable performance.

## 3.2 ImageNet

Orthogonal-SGDM also works on a large data set such as ImageNet (Deng et al., 2009) — Figure 6. Using a resnet34, mini-batch size of 1024, learning rate of $10^{-2}$, momentum of 0.9, and a weight decay of $5 \times 10^{-4}$, for 100 epochs. SGDM achieves a test accuracy of 61.9% and a test loss of 1.565 while Orthogonal-SGDM achieves 67.5% and 1.383 respectively. While these results are some way off the capabilities of the model

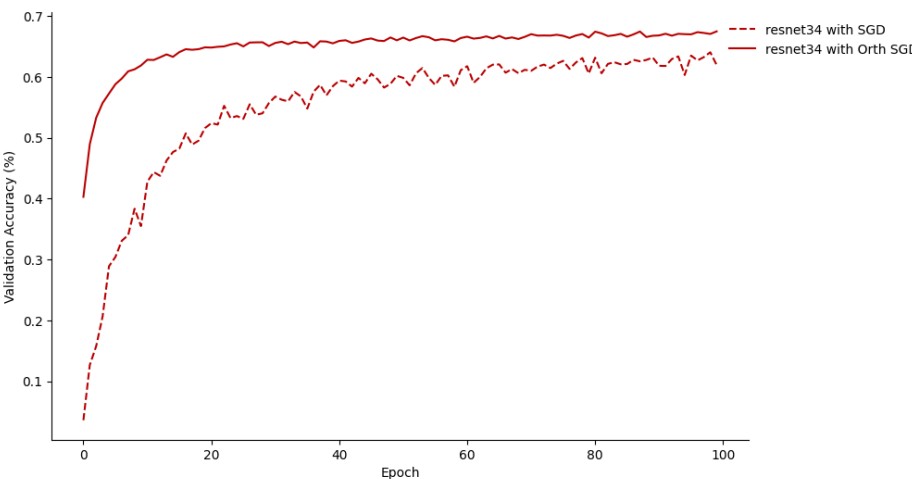

Figure 6: Validation accuracy of SGDM vs Orthogonal-SGDM on ImageNet

they still demonstrate a significant speed-up and improvement from using Orthogonal-SGDM, especially at the start of learning, comparable to the CIFAR-10 results.

## 3.3 Barlow Twins

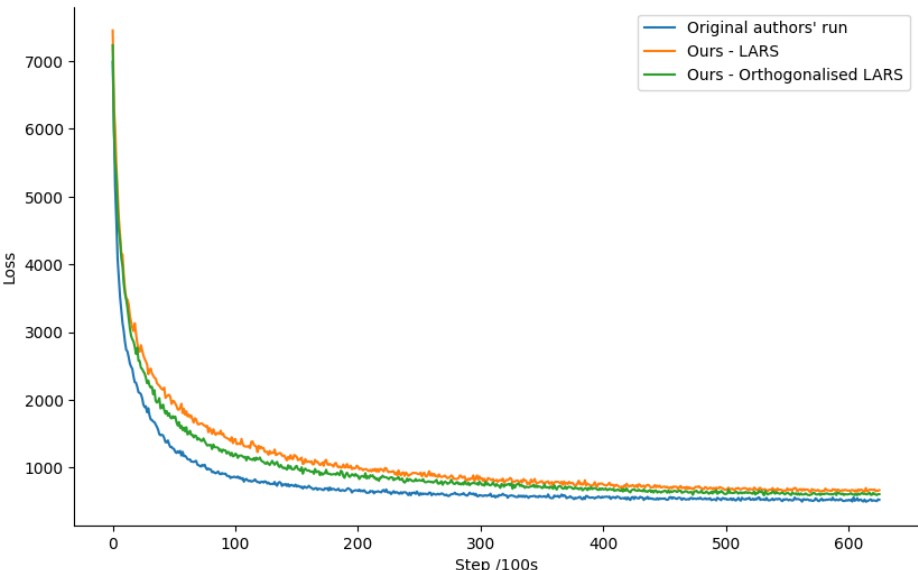

Figure 7: Barlow Twins loss during the unsupervised phase using LARS and Orthogonal-LARS on ImageNet

Barlow Twins (Zbontar et al., 2021) is a semi-supervised method that uses "the cross-correlation matrix between the outputs of two identical networks fed with distorted versions of a sample" to avoid collapsing to trivial solutions. While the authors do provide code, we could not replicate their results by running it. To train within our compute limitations we used a mini-batch size of 1024 instead of 2048 however this should not affect the results since "Barlow Twins does not require large batches" (Zbontar et al., 2021). Additionally, Barlow Twins uses the Layer-wise Adaptive Rate Scaling (LARS) algorithm (You et al., 2017), which is designed to adjust the learning rate based on the ratio between the magnitudes of the gradients and weights, there should be no significant slow-down, or speed-up, in learning due to the mini-batch size. We do not orthogonalise the gradients for the dense layers (see Section 4.4.1).

Comparing our own runs, we establish that orthogonalising the gradients before the LARS algorithm does speed up learning as shown in Figure 7, in agreement with previous experiments. This is evidence that orthogonalising gradients is also beneficial for semi-supervised learning and, moreover, that most first-order optimisation algorithms can be improved in this way.

## 4 Analysis of the method

### 4.1 Gradient normalisation

When we perform SVD on the reshaped gradient tensor, we obtain an orthonormal matrix, since this changes the magnitude of the vector we look at the effect of this normalisation. Normalised SGDM (N-SGDM) (Nesterov, 2003) provides an improvement in non-convex optimisation since it is difficult to get stuck in a local minimum as the step size is not dependent on the gradient magnitude. However, it hinders convergence to a global minimum since there is no way of shortening the step size; however, deep learning is highly non-convex and is unlikely to be optimised to a global minimum. Therefore, it stands to reason that normalising the gradient would speed up the optimisation of deep networks.

In addition, we compare N-SGDM to normalising the gradients per component — *i.e.* normalising the columns of $G_l$, Equation (3) without of orthogonalising them — Component Normalised SGDM (CN-SGDM). N-SGDM improves over SGDM, and CN-SGDM might improve over N-SGDM except for the oft-case where it diverges. Finally, Orthogonal-SGDM obtains the best solutions while remaining stable on all the models.

Table 4: Test accuracy for several models trained with SGDM, Normalised SGDM, Component Normalised SGDM, and Orthogonal-SGDM; trained as in Section 3.1.3.

|  | SGDM | N-SGDM | CN-SGDM | Orthogonal-SGDM |
|---|---|---|---|---|
| BasicCNN | $73.68_{\pm 0.27}$ | $73.72_{\pm 0.45}$ | $74.53_{\pm 0.32}$ | $\mathbf{76.75}_{\pm 0.23}$ |
| resnet18 | $76.83_{\pm 0.22}$ | $78.94_{\pm 0.19}$ | $0.00_{\pm 0.00}$ | $\mathbf{84.94}_{\pm 0.10}$ |
| resnet50 | $69.35_{\pm 0.30}$ | $79.35_{\pm 0.21}$ | $0.00_{\pm 0.00}$ | $\mathbf{86.59}_{\pm 0.10}$ |
| resnet44 | $79.73_{\pm 1.27}$ | $83.60_{\pm 0.77}$ | $84.44_{\pm 0.55}$ | $\mathbf{87.49}_{\pm 0.39}$ |
| densenet121 | $75.45_{\pm 0.20}$ | $79.06_{\pm 0.04}$ | $0.00_{\pm 0.00}$ | $\mathbf{84.86}_{\pm 0.07}$ |

### 4.2 Diversified intermediary representations

Along with different parametrisations we also desire different intermediary latent features, a model will perform better if its layers output $N$ different representations as opposed to $N$ similar ones.

Let $x_l$ be the resulting representations from the intermediary layers,

$$x_l = \left( \circ_{i=1}^l f_i \right) (x_0)$$

where $x_0$ is the input and $x_l$ is the intermediary representation after layer $l$. And thus $x_{li}$ the representation provided by $c_{li}$.

We look at the statistics of the absolute cosine of all distinct pairs of different representations,

$$R_l = \left\{ \left| \langle x_{li}, x_{lj} \rangle_2 \right| \mid i < j \right\}.$$

and see that the representations have smaller cosines, *i.e.* they are more diverse, when using Orthogonal-SGDM versus SGDM — Figure 8. In addition, Orthogonal-SGDM shows a decline in cosine similarity throughout training while SGDM does not; this is likely due to Orthogonal-SGDM having a higher regularisation effect and indicates that more information is being passed to the next layer as the network is trained.

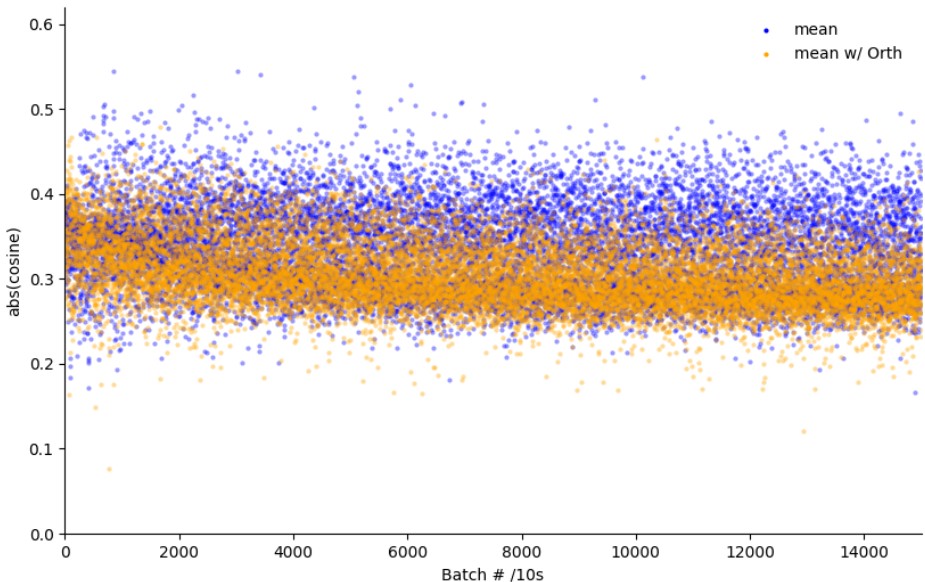

Figure 8: Mean of the absolute cosine of all distinct pairs of different intermediary representations, $\mathbb{E}\left[R_l\right]$, $l \in \{1, 2, 3\}$, for all layers of a BasicCNN trained on CIFAR-10 as in Section 3.1.3.

### 4.3 Disabled parameters

Disabled parameters occur when the activation function has a part with zero gradient, *e.g.* a Rectified Linear Unit (ReLU). If the result of the activation remains in this part, then the gradients of the preceding parameters will be zero and prevented from learning. This can limit the model's capacity, in opposition, however, Temporarily Disabled Parameters (TDP), *i.e.* parameters whose activation function doesn't "activate" for a single batch, can be beneficial and act as a regulariser, similar to dropout. To detect TDP, we simply look for parameters with zero gradient. Comparing the amount of TDP produced by SGDM versus Orthogonal-SGDM, Figures 9a and 9b respectively, shows that Orthogonal-SGDM ends with around and order of magnitude more TDP. This implies a much higher regularisation which helps to explain Orthogonal-SGDM's insensitivity to over-parametrisation.

### 4.4 Decomposition implementation

While QR decomposition is the most common orthogonalisation method, here it is, in practice, less stable than SVD as the gradients are rank deficient (Demmel, 1997, Section 3.5), *i.e.* they have at least one small singular value. Orthogonal-SGDM has a longer wall time than SGDM because of the added expense of the SVD which has non-linear time complexity in the matrix size. In practice, we have found that the calculation of SVD is either more than made up for by the speed up in iterates or a prohibitively expensive cost, with dense layers being the largest and thus most problematic.

While there exist methods for computing an approximate SVD which are faster, we have used PyTorch's default implementation since we are more concerned with Orthogonal-SGDM's performance and efficiency in iterates and not in wall time. Even so the overhead is small, training a resnet20 as in Section 3.1.3 takes 720.3 seconds with 96.4 of them taken up by the SVD calculation — an increase of 15.5% over normal SGDM. While this is a significant amount of time we can see that our method can take less than 5% of the number of epochs to reach the same accuracy — Figure 2.

It is doubtful that convergence of SVD is needed, so a custom matrix orthogonalisation algorithm, that has the required stability but remains fast and approximate, will reduce the computation overhead significantly and may allow previously infeasible networks to be optimised using Orthogonal-SGDM. However, we note

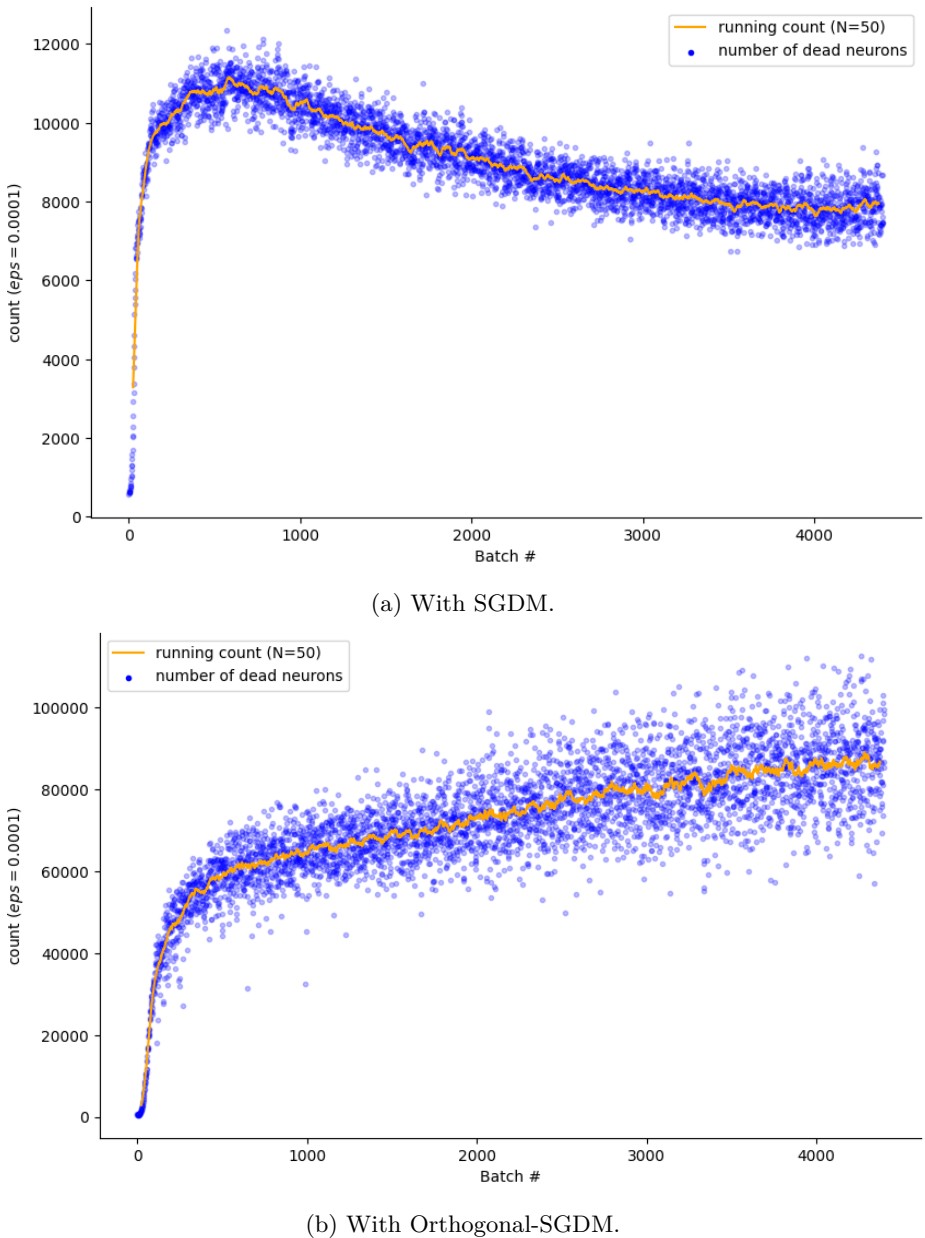

(a) With SGDM.

(b) With Orthogonal-SGDM.

Figure 9: Number of TDP for a the layer "layer2[1].conv2" of a resnet50 trained as in Section 3.1.3.

that even with this more suitable implementation, the runtime cost of this method would still bias towards many smaller layers for a deeper, thinner network.

### 4.4.1 Fully connected layers

Fully connected or dense layers also fit our component model from Equation (2) where the components are based on the inner product of the input and the parametrisation,

$$c_{l_i}(x) = \sigma(\langle \text{flatten}(x), \theta_{l_i} \rangle),$$

where $\sigma$ is an activation function, $S_l = 1$ giving $f_l : \mathbb{R}^{S_{l-1} \times N_{l-1}} \to \mathbb{R}^{N_l}$ and $\theta_l \in \mathbb{R}^{S_{l-1} \cdot N_{l-1} \times N_l}$ as desired (Wang et al., 2020). Intuitively, each column of the weight matrix acts as a linear map resulting in one item in the output vector. Thus, the gradients of fully connected layers can also be orthogonalised.

Table 5: Test accuracies and losses for Orthogonal-SGDM on CIFAR-10 when orthogonalising all layers vs orthogonalising just the convolutional layers. Trained as in Section 3.1.3, standard error across five runs.

| Accuracy | SGDM | Orthogonal-SGDM | Conv Orthogonal-SGDM |
|---|---|---|---|
| BasicCNN | $73.60_{\pm 0.19}$ | $76.67_{\pm 0.10}$ | $\mathbf{76.80}_{\pm 0.18}$ |
| resnet34 | $75.86_{\pm 0.26}$ | $85.42_{\pm 0.33}$ | $\mathbf{85.68}_{\pm 0.21}$ |
| resnet20 | $79.14_{\pm 0.62}$ | $87.12_{\pm 0.12}$ | $\mathbf{87.70}_{\pm 0.40}$ |
| Loss | SGDM | Orthogonal-SGDM | Conv Orthogonal-SGDM |
| BasicCNN | $0.7603_{\pm 0.0061}$ | $0.6808_{\pm 0.0038}$ | $0.6732_{\pm 0.0041}$ |
| resnet34 | $1.0468_{\pm 0.0134}$ | $0.7087_{\pm 0.0165}$ | $0.6268_{\pm 0.0105}$ |
| resnet20 | $0.6728_{\pm 0.0301}$ | $0.6766_{\pm 0.0155}$ | $0.4824_{\pm 0.0225}$ |

As noted above, Section 4.4, the extra wall time is dominated by the largest parameter, this is often the final dense layer; Table 5 shows that, for CIFAR-10, only orthogonalising the convolutional layers does not reduce performance. While both the error rates actually decrease when for Conv Orthogonal-SGDM we hesitate to say that orthogonalising dense layers is detrimental since the results are within the margin of experimental error; additionally these networks only have a dense final classification layer which is qualitatively different from intermediary dense layers, and only making up a small proportion of the parameters.

### 4.5 Limitations with a small mini-batch size

Orthogonal-SGDM does not perform as well as SGDM when the mini-batch size is extremely small, Figure 10, due to the increased levels of noise in the gradients which is compounded during SVD.

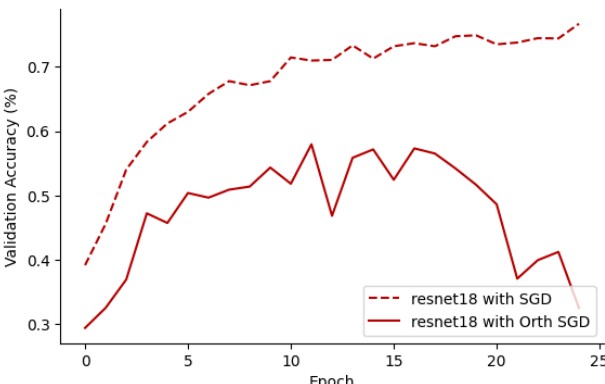

Figure 10: CIFAR-10 with mini-batch size=4 trained as in Section 3.1.3.

Orthogonal-SGDM starts to outperform SGDM with a mini-batch size of 16 on a resnet18 for CIFAR-10. Few models need such small mini-batch sizes, but if they do then SGDM would be a more suitable optimisation algorithm. In addition to the learning collapse, the time taken by SVD is only dependent on the parameter size and not the mini-batch size, as the gradients are accumulated before the SVD, so increasing the number of mini-batches per epoch also increases the wall time to train.

## 5 Conclusion

In this work we have laid out a new optimisation method, tested it on different models and data sets, showing state-of-the-art results and out of the box robustness to hyper-parameter choice and over-parametrised models. Not only is Orthogonal-SGDM better for image classification it also has practical application in

problems such as object detection and semantic segmentation since they make use of a pre-trained image classification backbone.

Lastly, we mentioned briefly in Section 1 how attention heads fit our component model but, since they are beyond the scope of this work, we will explore the potential gain in using Orthogonal-SGDM with them in a future work, and expect a similarly gain will be obtained. Using Orthogonal-SGDM to train language models is a promising avenue for future work, as it is a very computationally expensive task.

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

# A  Model Summaries

## A.1  BasicCNN

| Layer (type) | Output Shape | Param # |
|---|---|---|
| Conv2d−1 | [−1, 32, 16, 16] | 896 |
| BatchNorm2d−2 | [−1, 32, 16, 16] | 64 |
| Conv2d−3 | [−1, 32, 8, 8] | 9,248 |
| BatchNorm2d−4 | [−1, 32, 8, 8] | 64 |
| Conv2d−5 | [−1, 32, 4, 4] | 9,248 |
| BatchNorm2d−6 | [−1, 32, 4, 4] | 64 |
| Linear−7 | [−1, 10] | 5,130 |
| BasicCNN−8 | [−1, 10] | 0 |

Total params: 24,714
Trainable params: 24,714
Non−trainable params: 0

Input size (MB): 0.01
Forward/backward pass size (MB): 0.16
Params size (MB): 0.09
Estimated Total Size (MB): 0.27

## B Full results plot

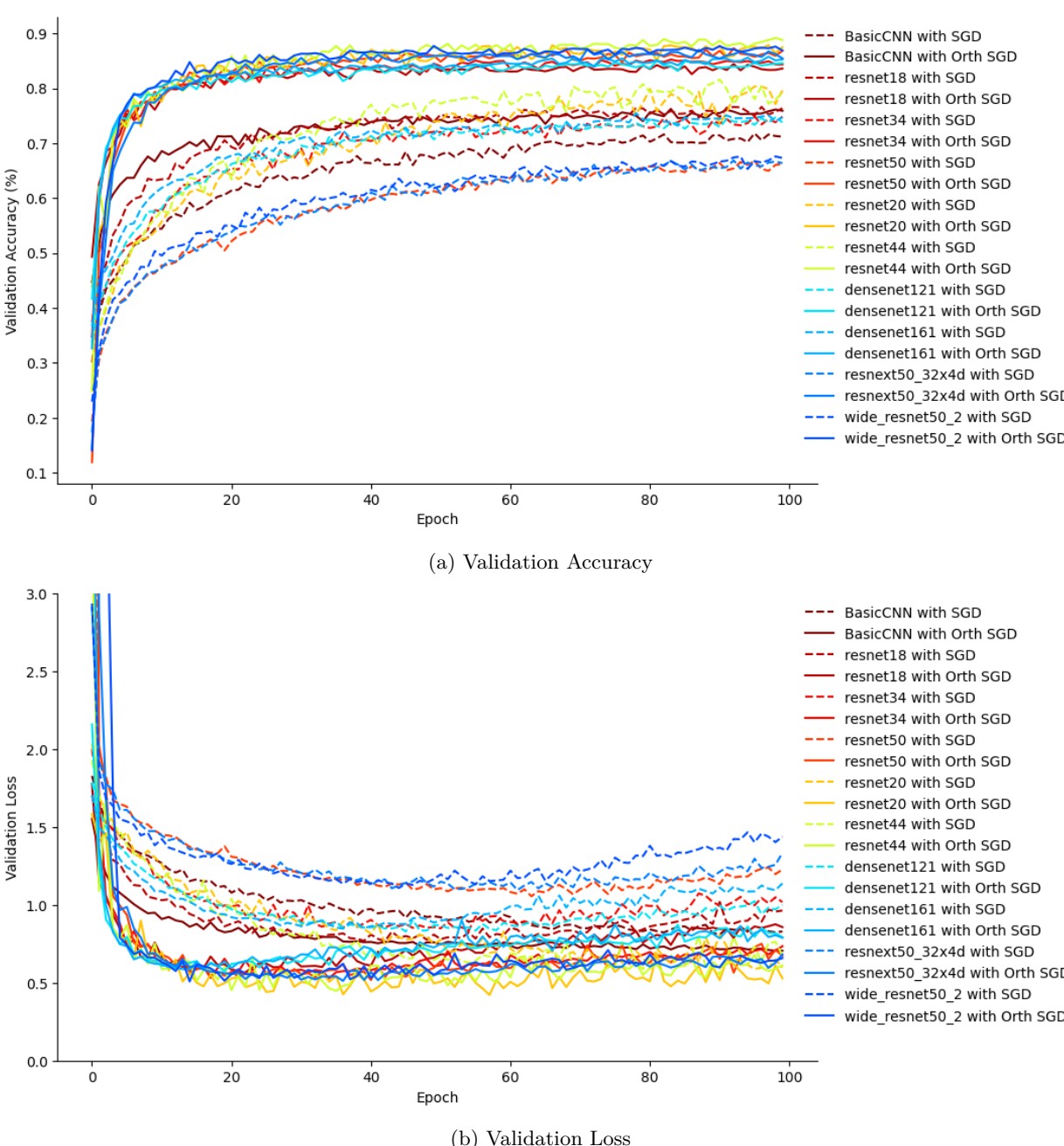

(a) Validation Accuracy

(b) Validation Loss

Figure 11: SGDM vs Orthogonal SGDM

Table 6: Test loss across a suite of hyper-parameter sets on CIFAR-10 on a resnet20, standard error across five runs. For Adam $\beta_2 = 0.99$.

| | | SGDM | | Adam | |
|---|---|---|---|---|---|
| LR | Momentum | Original | Orthogonal | Original | Orthogonal |
| $10^{-1}$ | 0.95 | $0.5487_{\pm 0.0913}$ | $\mathbf{0.4916}_{\pm 0.0420}$ | $2.4148_{\pm 0.7090}$ | $0.7938_{\pm 0.0663}$ |
| $10^{-2}$ | 0.95 | $0.5762_{\pm 0.0431}$ | $\mathbf{0.4772}_{\pm 0.0277}$ | $0.8395_{\pm 0.1079}$ | $0.5415_{\pm 0.0153}$ |
| $10^{-3}$ | 0.95 | $0.9323_{\pm 0.0149}$ | $\mathbf{0.4988}_{\pm 0.0114}$ | $0.6013_{\pm 0.0305}$ | $0.5998_{\pm 0.0131}$ |
| $10^{-1}$ | 0.9 | $0.6089_{\pm 0.0535}$ | $\mathbf{0.5370}_{\pm 0.0230}$ | $9.0193_{\pm 5.9607}$ | $0.9803_{\pm 0.1607}$ |
| $10^{-2}$ | 0.9 | $0.6217_{\pm 0.0194}$ | $\mathbf{0.4841}_{\pm 0.0087}$ | $0.8898_{\pm 0.0447}$ | $0.5903_{\pm 0.0201}$ |
| $10^{-3}$ | 0.9 | $1.0977_{\pm 0.0036}$ | $\mathbf{0.5042}_{\pm 0.0055}$ | $0.6045_{\pm 0.0312}$ | $0.5829_{\pm 0.0102}$ |
| $10^{-1}$ | 0.8 | $\mathbf{0.5395}_{\pm 0.0091}$ | $0.5414_{\pm 0.0459}$ | $4.5143_{\pm 1.6628}$ | $1.0984_{\pm 0.2710}$ |
| $10^{-2}$ | 0.8 | $0.6669_{\pm 0.0290}$ | $\mathbf{0.4940}_{\pm 0.0163}$ | $0.8741_{\pm 0.0627}$ | $0.5114_{\pm 0.0204}$ |
| $10^{-3}$ | 0.8 | $1.2805_{\pm 0.0106}$ | $\mathbf{0.5238}_{\pm 0.0034}$ | $0.6950_{\pm 0.0656}$ | $0.5671_{\pm 0.0088}$ |
| $10^{-1}$ | 0.5 | $0.6796_{\pm 0.0158}$ | $\mathbf{0.5003}_{\pm 0.0148}$ | $15683.8291_{\pm 15681.5332}$ | $0.9481_{\pm 0.1072}$ |
| $10^{-2}$ | 0.5 | $0.8927_{\pm 0.0304}$ | $\mathbf{0.4950}_{\pm 0.0121}$ | $0.9594_{\pm 0.0846}$ | $0.6610_{\pm 0.0285}$ |
| $10^{-3}$ | 0.5 | $1.5309_{\pm 0.0189}$ | $0.6284_{\pm 0.0186}$ | $0.6482_{\pm 0.0511}$ | $\mathbf{0.6228}_{\pm 0.0213}$ |
| $10^{-1}$ | 0 | $0.7587_{\pm 0.0354}$ | $\mathbf{0.6298}_{\pm 0.0225}$ | $2.3370_{\pm 0.0310}$ | $1.6441_{\pm 0.4466}$ |
| $10^{-2}$ | 0 | $1.2765_{\pm 0.0076}$ | $\mathbf{0.5960}_{\pm 0.0235}$ | $1.4223_{\pm 0.2461}$ | $0.7786_{\pm 0.0741}$ |
| $10^{-3}$ | 0 | $1.8124_{\pm 0.0030}$ | $1.0297_{\pm 0.0150}$ | $0.7565_{\pm 0.0905}$ | $\mathbf{0.5854}_{\pm 0.0285}$ |

Table 7: Test loss for several models trained with SGDM, Normalised SGDM, Component Normalised SGDM, and Orthogonal-SGDM.

| | SGDM | N-SGDM | CN-SGDM | Orthogonal-SGDM |
|---|---|---|---|---|
| BasicCNN | $0.7559_{\pm 0.0065}$ | $0.7637_{\pm 0.0098}$ | $0.7443_{\pm 0.0094}$ | $\mathbf{0.6824}_{\pm 0.0081}$ |
| resnet18 | $0.9252_{\pm 0.0098}$ | $0.9726_{\pm 0.0214}$ | $\text{nan}_{\pm \text{nan}}$ | $\mathbf{0.7938}_{\pm 0.0083}$ |
| resnet50 | $1.0950_{\pm 0.0181}$ | $0.9454_{\pm 0.0126}$ | $\text{nan}_{\pm \text{nan}}$ | $\mathbf{0.6785}_{\pm 0.0076}$ |
| resnet44 | $\mathbf{0.7093}_{\pm 0.0678}$ | $0.7641_{\pm 0.0441}$ | $0.7902_{\pm 0.0477}$ | $0.7694_{\pm 0.0426}$ |
| densenet121 | $0.9357_{\pm 0.0071}$ | $1.0096_{\pm 0.0167}$ | $\text{nan}_{\pm \text{nan}}$ | $\mathbf{0.8142}_{\pm 0.0084}$ |

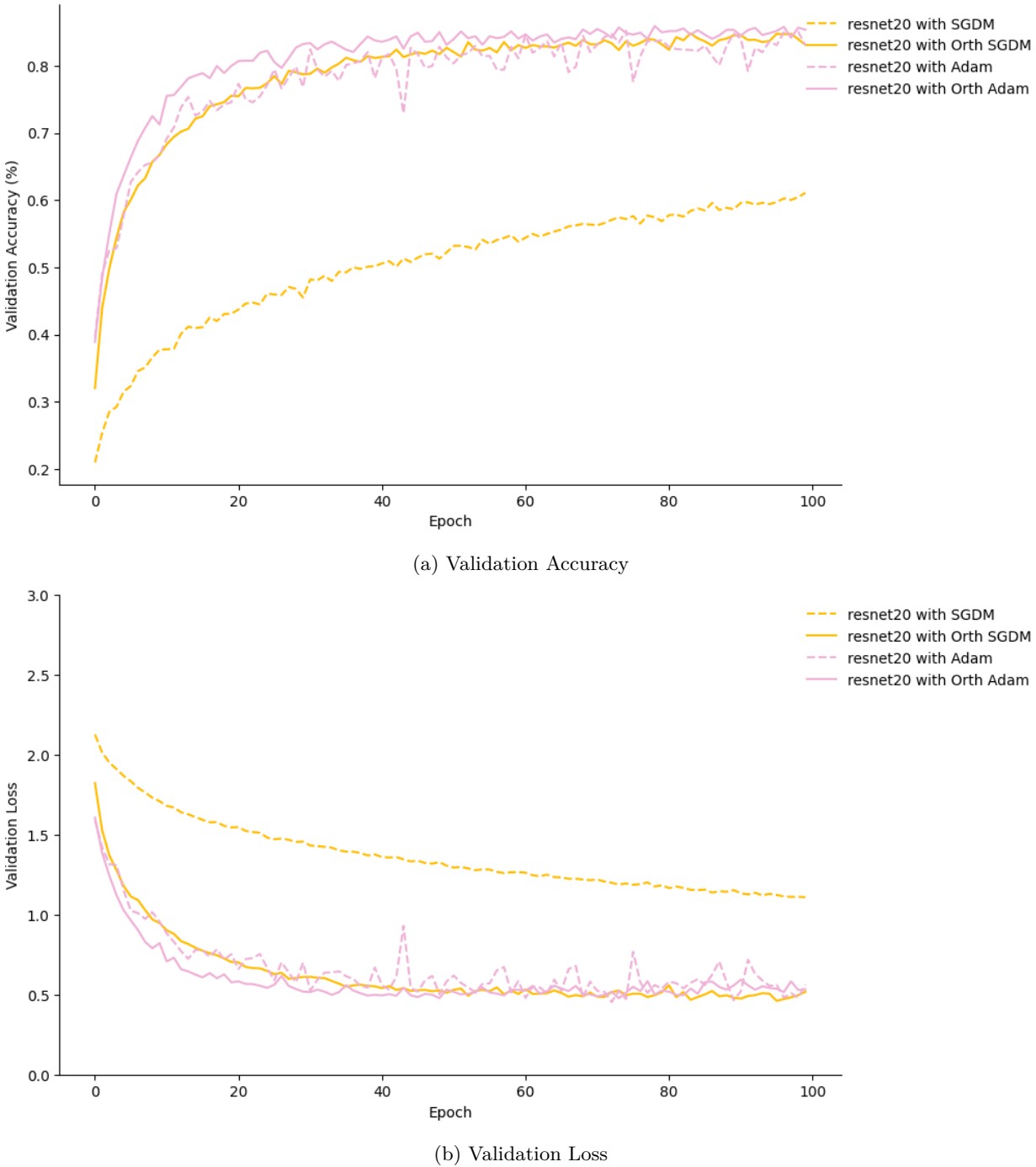

(a) Validation Accuracy

(b) Validation Loss

Figure 12: A compassion of Adam and SGDM, learning rate $= 1 \times 10^{-3}$, $\beta_1 = 0.9$, $\beta_2 = 0.99$.

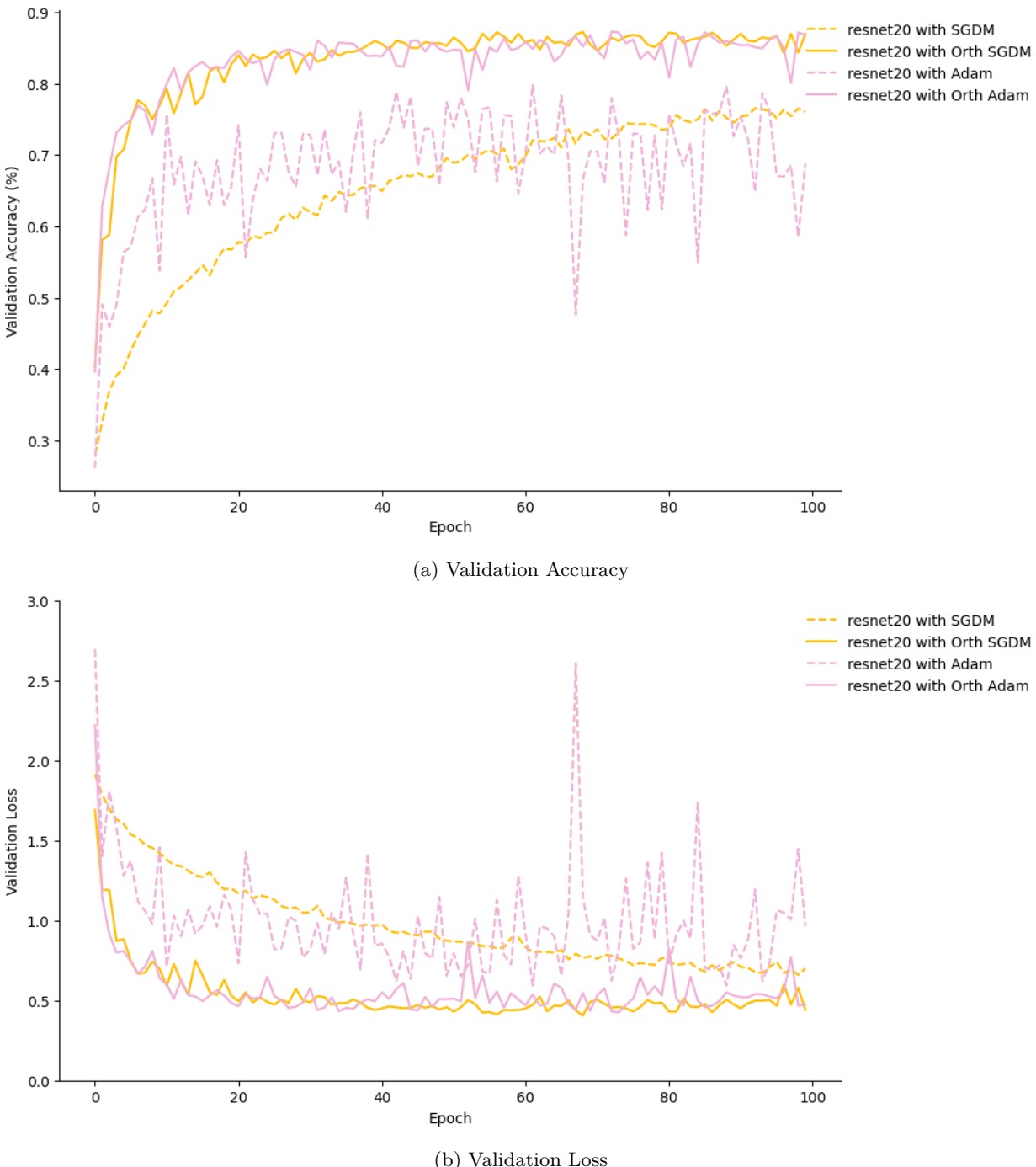

(a) Validation Accuracy

(b) Validation Loss

Figure 13: A compassion of Adam and SGDM, learning rate $= 1 \times 10^{-2}$, $\beta_1 = 0.8$, $\beta_2 = 0.99$.

