# OpenReview forum: "Orthonormalising gradients improves neural network optimisation"
_TMLR — Withdrawn by Authors_

### Review · Reviewer_4TBh · 2023-05-18

**Summary Of Contributions:**

This work proposes to orthogonalize gradients updates at each layer to both improve performance as well as convergence of image classification models.

**Audience:**

No

**Claims And Evidence:**

No

**Requested Changes:**

It is my current opinion that this work has fatal flaws. Notably, I do not believe it will converge in the convex case, even when mild conditions, such as strictly convex and strict smoothness, are assumed (see A1-A4 in the Appendix of the PCGrad paper for the proofs).


Moreover
```
We do this by orthogonalising the gradients before they are used by an optimisation method rather than modifying the weights themselves. This, in effect, biases the training towards learning orthogonal representations and we show this to be the case; in many areas of deep learning, introducing a bias instead of a hard constraint is preferred.
```
seems to be the main thrust of the author's arguments behind orthogonalizing gradients.

There is a clear link between orthogonalizing parameters and orthogonal representations; however, I do not see a direct link between orthogonalizing gradients and orthoognalizing learned representations when orthogonalizing gradients does not necessarily lead to orthogonal parameters.

I believe even the notion of learning orthogonal representations would be highly contested as unilaterally beneficial without substantive theoretical analysis and experimental results. Especially for vision where early-layer filters appear to learn edge detectors (Figure 3 of http://www.cs.toronto.edu/~fritz/absps/imagenet.pdf), forcing orthognality into the learning dynamics would most likely be detrimental.

**Strengths And Weaknesses:**

Strengths:
1. The empirical findings are quite strong.

Weaknesses:
1. I believe orthogonalizing gradients will not converge in the convex case, even when mild conditions are assumed (strictly convex, strictly smooth). This is a fatal flaw.
2. Orthogonalizing gradients at each update step does not necessarily orthogonalize the parameters. In fact, it can lead to stationary parameters at non-stationary points (one can effectively create "saddle points" where none existed before).
3. PCGrad doesn't quite orthogonalize gradients (beginning of related work at the end of page 1), it would be more precise to say they project opposite directions into the normal plane of one of the gradients (sampled uniformly at random from the set of gradient vectors) -- see Algorithm 1 of the PCGrad paper.

---

> ### Author Response · Authors · 2023-06-08
> **Thank you for the reivew**
>
> Thank you for taking an interest in our paper, and your helpful comments in your review. The criticism about the algorithm not converging is interesting and perhaps deserves a discussion (obviously having pages limits does not help).  We agree that because we normalize the gradients, our algorithm will not converge in a quadratic minimum.  The question is whether this is a fatal flaw.  Clearly, from the perspective of classical optimisation, this seems very strange.  However, training deep networks is, in our view, very different from most classical optimisation problems.  In particular, given the size of the search space, we do not believe that we are ever near to converging (we are not even sure what convergence means when we use mini-batches each with its own optimum).  As we explained in addressing the previous referee, there are many reasons we believe that convergence is not particularly relevant.  Even using conjugate gradient, the number of updates needed to converge to a quadratic minimum would be equal to the dimension of the search space, which for deep learning often would be of the order greater than 10^7.  In practice, the number of updates used is orders of magnitude less than this.  Even the momentum decay of 0.9 is sufficiently high that it seems unlikely that we would be close to convergence after a few hundred epochs.  Moreover, it is not clear that convergence would even by desirable as early stopping is a well-known regularization strategy (we would argue that it is not done explicitly in deep learning regularly because networks are never close to converging).  Thus, the classical notion of converging seems to us not to be a fatal flaw.  We could improve the convergence of our algorithm on a quadratic by not normalizing the gradients, but empirically that leads to a much slower learning on deep networks.  In our view, we would prefer to use an algorithm that empirically works well than a worse algorithm that has a guaranteed convergence on a simple problem when we know that we will never get close to convergence in practice.
>
> Even so, building a model showing the advantages of our approach is a much subtler task than it may appear.  It is not clear how orthogonalising channel gradients would present a quadratic minimum.  The idea is fairly simple, when performing some task, there are usually a number of features that are important, but some of these features will be more important than others.  In deep neural networks with identical channels, there is the potential for many channels to learn the same significant feature, but it is a waste of resources for these channels to learn redundant information.  The aim of orthogonalizing the gradient is to encourage the channels to learn different features.  We believe it would be possible to build an idealized model that would demonstrate the advantage of our approach, but it would be open to criticism that it was an artificial model designed to favour our approach (which of course it would be).  We believe that the empirical evidence of its benefits is more persuasive.
>
> It is true that orthogonalising gradients does not necessarily (and almost surely doesn't) orthogonalise the parameters; however, other work has shown that "Training to obtain a strict [orthogonal parametrisation] amounts to optimizing the weight matrices over their respective Stiefel manifolds, which, however, is very costly for large-sized DNNs" [1] and that it is not worth the computation and time: "approximate [orthogonal parametrisations] perform as well as strict ones, but at a much lower computational cost" [1]
>
> While orthogonalising the gradients can lead to a zero update where the gradient is non-zero, this is, in practice, avoided by the gradient normalisation and momentum.
>
> PCGrad takes a gradient for one task and orthogonalises it (via a Gram–Schmidt projection) with all other gradients which are conflicting. Since this is done independently for each task, it is true that ${g^{PC}_i} $ does not form an orthogonal set, however each $g^{PC}_i$ is formed by performing orthogonalisation on gradients (except in the case where a task has no conflicting task gradients).
>
> [1] Jia et al. Orthogonal deep neural networks. IEEE transactions on pattern analysis and machine intelligence, 2019

---

### Review · Reviewer_HSW3 · 2023-05-28

**Summary Of Contributions:**

This manuscript proposes to modify the standard stochastic optimization algorithms in deep learning by projecting the gradient vectors of each layer towards their nearest orthogonal matrix before the update step. Throughout the paper, a series of experiments aim to substantiate the hypothesis that the orthogonally modified versions of popular optimizers, such as SGD, ADAM, or LARS confer substantial benefits over their traditional, unrestricted counterparts. The enhancements are claimed in terms of both training speed and test accuracy across various benchmark tasks.

Further, the authors postulate that these orthogonally revised algorithms exhibit a reduced sensitivity to hyperparameter adjustments and are optimally compatible with non-convex optimization tasks as they are not sensitive to scale and because "deep learning is unlikely to be optimized to a global minimum". Other experiments in the manuscript seek to imply that these orthogonal optimizers yield more diverse feature representations. The authors draw attention to a modest decrease in the mean cosine similarity among features when training a basic CNN with orthogonal-SGD, as contrasted with its training using conventional SGD.

Finally, the authors discuss the limitations of their proposed algorithms and highlight its poorer performance with small mini-batches and increase in runtime. According to the authors, none of these limitations are significant in practice.

**Audience:**

No

**Broader Impact Concerns:**

I see no clear societal concerns stemming from this work.

**Claims And Evidence:**

No

**Requested Changes:**

Regrettably, I believe that this manuscript cannot be sufficiently improved within the scope of reasonable revisions, hence my recommendation will very likely be rejection regardless of any changes.

Should the authors consider resubmitting this work in the future, I would expect a substantial revamp of all experimental sections to ensure a fair comparison among different techniques. All optimizers should be assessed under the same fair settings and be tuned with the same budgets. This would allow to draw meaningful conclusions about their relative performance and hyperparameter sensitivity.

Additionally, the proposed method requires clearer motivation, supplemented by a deeper and more critical evaluation of its dynamics. For example, any claims regarding superior feature representations should be substantiated with measurable improvements in downstream task performance. Some theoretical analysis on simple settings with reasonable assumptions that can help clarify the intuitions of this technique would also be an important asset.

Lastly, the paper needs to be better situated within the existing body of research, acknowledging and building upon relevant literature in the field.


**Strengths And Weaknesses:**

## Strengths

#### 1. **Importance of the research area**:

Optimization is a critical part of deep learning. Any new ideas that can improve how we do optimization in deep learning have the chance to make a big difference in the field.

#### 2. **Interesting idea**:

As far as I know, the idea of making the updates of standard deep learning optimizers orthogonal is new. The paper doesn't explain very well why this would help optimization, but it's still worth exploring what effects it could have on training and testing performance.

## Weaknesses

#### 1. (Major) **Unfair comparison of different optimizers**:

The main problem with this work is that it doesn't compare different optimizers in a fair way. All the results in the paper, except for the ones in Table 1, come from settings where the optimizers don't perform their best. For instance, the test accuracies on CIFAR10 in Table 2 and Table 3 are much lower than what can be achieved with standard optimizers and the architectures used. This also applies to the results this paper reports for the orthogonal optimizers (as compared to the "best" results provided in Table 1). Therefore, we can't trust the conclusions drawn from these experiments.

Overall, the benchmarking practice in this work is of very low quality and can be misleading for the community. It seems to cherry-pick results that make the orthogonal optimizers look better, and this isn't acceptable in a serious and scientific journal like TMLR.

Even Table 1, which supposedly shows the best results for all the optimizers, is also flawed because it only shows results where the Orthogonal-SGDM has been tuned using Bayesian optimization. To be fair, the table should also show results of SGDM tuned with the same Bayesian strategy used for Orthogonal-SGDM. The paper also doesn't explain how this tuning strategy was designed anywhere in the text.

#### 2. (Major) **Poor motivation for the technique and lack of thorough analysis**:

Even though it could be a potentially interesting idea, the paper doesn't explain very well why making the updates orthogonal would be a good thing. For example, it doesn't explain why this change should improve convergence or generalization. Also, it makes some wrong claims, like saying that most optimization in deep learning doesn't reach global optima. This is false, as in many overparameterized cases the loss gets very close to zero and training accuracy reaches 100%.

Furthermore, the paper doesn't offer a deep and serious analysis of how the orthogonal optimizers work. Sections 4.2 and 4.3 provide almost no useful information about why these optimizers should work. They use metrics that are flawed and hard to understand. For example, a small decrease in cosine similarity in such high dimensional spaces is almost meaningless, and even if it wasn't, it's unclear why we should care about this metric. If we want to measure the quality of a representation, we should test how it performs in some downstream task, such as transfer learning, out-of-distribution detection, or guiding generative models.

#### 3. **Poor presentation**:

The writing is hurried, the math is poorly formatted, and the structure of the paper is confusing. The tone of the paper is also not scientific enough and lacks rigor. In general, the paper needs a lot of work to improve its writing and attention to detail. For example, the paper wrongly refers to Barlow Twins as a semi-supervised learning method, when it is actually a self-supervised method.

#### 4. **Weak discussion of related work**:

The paper fails to recognize many past studies that have explored theoretically and practically the use of orthogonal weights at initialization and during training. Here are some examples, but it would be good to also check out their references:
- Trockman, A. & Kolter, J. Z. Orthogonalizing Convolutional Layers with the Cayley Transform. ICLR, 2021.
- Xiao, L., Bahri, Y., Sohl-Dickstein, J., Schoenholz, S. S., & Pennington, J. Dynamical Isometry and a Mean Field Theory of CNNs: How to Train 10,000-Layer Vanilla Convolutional Neural Networks. ICML, 2018.
- Arjovsky, M., Shah, A., & Bengio, Y. Unitary evolution recurrent neural networks. ICML, 2016.

By not fully acknowledging related work, the paper lacks context. Looking at the relevant literature could maybe help improve the motivation of this work.

---

> ### Author Response · Authors · 2023-06-08
> **Thank you for the reivew**
>
> The test accuracies in Table 2 & 3 are lower than the state-of-the-art since we wanted to show the optimisers over a range of values rather than the singular settings where they perform their best (which is Table 1) as this would make it appear as if we only reported our best runs. We acknowledge your comment on cherry-picking, this was not our intention as we have reported the results from all the experiments we ran except the Bayesian optimisation on SGDM. Your comments on the Bayesian optimisation are valid and the experiment should have been included; this has been given attention in the next draft; the result of using Bayesian optimisation on SGDM was the only result left out because it underperformed relative to not only OSGDM but also to the original paper.
>
> We did not justify our claim that most optimisation in deep learning doesn't reach the global optimum because we felt this to be uncontroversial, and we have a page limit.  However, let us explain why we do not believe that in deep learning we have converged (even to a local optimum).  Firstly, we are not sure what convergence even means when using mini-batches as each mini-batch will have a different local optimum and we have only sampled a small proportion of all possible mini-batches.  We might assume that the learning rate is so small that we are effectively optimizing the loss for the whole training set, but this would suggest that the mini-batch size is irrelevant, which is usually not the case.
> Putting aside this problem. Secondly, in a quadratic minimum using a quasi-Newton method such as conjugate gradient the time to reach the minimum would be the dimensionality of the space we are searching (e.g. the number of parameters).  Given that typical deep learning models often have millions or even hundred of millions of parameters, then the number of updates would seem to be orders of magnitudes smaller than necessary for convergence in this ideal case.  Given that we're using a weaker optimiser and are not in a convex (or even everywhere differentiable) search space, we would be very puzzled at why we would converge.
> Thirdly, given that typically people use momentum with a decay rate of 0.9 it seems unlikely that the momentum buffer would have decayed to zero for the typical number of epochs being carried out.
> Fourthly, practitioners typically reduce their learning rate by a factor of 10 and find that the loss then rapidly decreases, suggesting that the parameters have not reached a minimum.
> Fifthly, empirically we have not observed a training loss curve where there is strong evidence that the gradient ever reaches zero (or even that it reduces at all, with it sometimes increasing when the learning rate is dropped. While it might decrease in size, in every example we have seen, it is hard to argue it is zero).  Furthermore, in our experience, when running any deep learning model for longer, we have always observed that the weights continue to change.  If you are aware of models where this does not happen, we would be interested.  In most instances, the training accuracy reaches 100% very rapidly (often after only a few epochs), but the loss continues falling for hundreds of epochs (as we said, we have never seen a case where the loss does not seem to be decreasing, albeit very slowly).  Correctly classifying the training set is not a criterion for converging or reaching the global minimum.  Furthermore, having a small loss (close to zero) is not, to the best of our knowledge, evidence of reaching a global minimum.  We therefore stand by our statement.  Obviously, if our arguments are incorrect, we would be truly interested to know, but from the comments made by you, we are rather puzzled by your assertion.
>
> Thank you for pointing out our typing mistake relating to the Barlow Twins' description.
>
> The first of the suggested papers is interesting and relevant, but was published after the majority of this work was done; however, it can be included at this stage. The latter papers are about orthogonal weights from the perspective that gradient norm preserving transforms are beneficial in regard to extremely deep networks and RNNs; since we do not constrain our weights to be orthogonal, this is not a property of the trained networks.

---

> > ### Comment · Reviewer_HSW3 · 2023-06-13
> > **Thank you for the response**
> >
> > Thank you for your reply. Unfortunately, after having read the authors' rebuttal and the other reviewers' comments, I stand by my previous assessment and I will recommend rejection of this work.
> >
> > In their rebuttal, the authors clearly position their work as an empirical contribution:
> >
> > > *[...] we would prefer to use an algorithm that empirically works well than a worse algorithm that has a guaranteed convergence on a simple problem when we know that we will never get close to convergence in practice.*
> >
> > In this regard, I would expect that the empirical evaluation of the proposed algorithm was solid, thorough, and as unbiased as possible. However, as I mentioned in my original review, the lack of proper tuning of all the tested algorithms in this work make the observations and comparisons presented in the manuscript meaningless. I acknowledge the comments of the authors in this regard, but I do not share their opinion: Not over-optimizing the performance of all the algorithms and showing ablations regarding stability to hyperparameter tuning is acceptable, but presenting results in very suboptimal training regimes that do not reflect the real use case of the algorithm as general is not scientifically correct.
> >
> > Furthermore, besides the lack of proper comparisons with prior work, and as commented by the other reviewers, the presented technique lacks motivation, deep analysis, and it is not well positioned with respect to the literature. For these reasons, I believe that this paper should not be accepted to TMLR as its claims cannot be verified, and the TMLR audience would not benefit from reading its findings.

---

### Review · Reviewer_XDcN · 2023-05-29

**Summary Of Contributions:**

The paper introduces an optimizer which encourages orthogonalized gradients between "components" of a deep network. Components can refer to different filters within a convolutional layer, or different rows of a weight matrix in a fully connected layer, or different attention heads of in a Transformer layer. Experiments are presented on CIFAR-10 and ImageNet for supervised learning, as well as some experiments on self-supervised learning. Some studies on "dead neurons" as well as weight orthogonality are presented.

**Audience:**

Yes

**Claims And Evidence:**

No

**Requested Changes:**

I would like to see this method applied to a simple case, for example the classical quadratic bowl used in many optimization papers. An analytical treatment would also be nice, can you show that the method converges?

**Strengths And Weaknesses:**

Strengths:
-method is simple and easy to implement
-results are seemingly good

Weakness:
-there is no analytical analysis whatsoever of this method, making it's properties difficult to understand
-the results seem suspect to me. For example why are the curves in Figure 1 & 6 separated at iteration 0 if they correspond to the same weight initialization (as they should).
-the proposed method is too computationally intensive to be used in many areas of interest

---

> ### Author Response · Authors · 2023-06-08
> **Thank you for the review**
>
> Thank you for your interest and your feedback on our paper. While analytical analysis is beneficial to understand how algorithms work, in our view, we would prefer to use an algorithm that empirically works well than one with guarantees on simple problems but is not shown to work, or performs worse in practice, for example, Adam is still preferred over AMSGrad; unfortunately we have a page limit, and so have preferred arguing the empirical case.
> Figures 1 & 6 are errors in the plots' axes labelling, the first data point is after epoch 1. The newly initialised model is random and has an accuracy approximately to 1 over the number of classes: 0.1 for these plots on CIFAR10.
> We comment in section 4.4 about the computational cost of our method, where, firstly, we find memory to be the predominant barrier to training larger models, and that the increase in time per iterate is made up for in the greater decrease in loss per iterate. Secondly, once we can establish that this method is functional and of interest, we can then look at finding approximations and better ways to orthogonalise the gradients, as has been done with orthogonalising weights previously.
> It is not clear how orthogonalising channel gradients would present a simple quadratic bowl minimum.  We believe it would be possible to build an idealized model that would demonstrate the advantage of our approach, but it would be open to criticism that it was an artificial model designed to favour our approach (which of course it would be).  We believe that the empirical evidence of its benefits is more persuasive.

---

> > ### Comment · Reviewer_XDcN · 2023-06-14
> > **Thank you for the response**
> >
> > I appreciate the motivation for valuing empirical performance, however I think the value of this work can be greatly increased if the authors included at least some analysis and experiments in simpler settings (e.g. quadratic objectives). As is often the case, insights gained from simplified settings often translate to more complex ones. Also I share the other reviewers concerns regarding fair comparisons to other methods.

---

### Public Comment · ~Alan_Jeffares1 · 2023-05-02
**Sharing some closely related prior work**

Dear Authors,

I enjoyed reading this interesting paper. I wanted to share our paper [1] which is closely related to this method but seems to have been missed in the related works. In this work we also propose a regularization term that orthogonalizes a neural network's gradients (however this work seems to have a distinct method for achieving this).  Our work also connects the effectiveness of this approach to the resulting diversity among neuron activations (we use ensemble theory to investigate this point).

However, I do think this submission contains significant novelty not considered in our previous work. This work considers new data modalities, self-supervised learning, and its integration into the optimization process. Therefore I believe this would be resolved by a comparison to [1] in the related work (although empirical comparisons would be interesting in addition).

Kind regards,
Alan Jeffares



[1] Jeffares, Alan, et al. "TANGOS: Regularizing Tabular Neural Networks through Gradient Orthogonalization and Specialization." The Eleventh International Conference on Learning Representations.

---

### Note · Authors · 2023-06-19

**Comment:**

The reviewers raise several objections that should be addressed before publication.

**Withdrawal Confirmation:**

I have read and agree with the venue's withdrawal policy on behalf of myself and my co-authors.